# *Mycoplasma genitalium* Provokes Seminal Inflammation among Infertile Males

**DOI:** 10.3390/ijms222413467

**Published:** 2021-12-15

**Authors:** Stanislav Tjagur, Reet Mändar, Olev Poolamets, Kristjan Pomm, Margus Punab

**Affiliations:** 1Andrology Centre, Tartu University Hospital, Ludvig Puusepa 8 Street, 50406 Tartu, Estonia; olev.poolamets@kliinikum.ee (O.P.); kristjan.pomm@kliinikum.ee (K.P.); margus.punab@kliinikum.ee (M.P.); 2Faculty of Medicine, University of Tartu, Ravila 19 Street, 50411 Tartu, Estonia; 3Department of Microbiology, Institute of Biomedicine and Translational Medicine, Faculty of Medicine, University of Tartu, Ravila 19 Street, 50411 Tartu, Estonia; 4Competence Centre on Health Technologies, Teaduspargi 13 Street, 50411 Tartu, Estonia; 5Institute of Clinical Medicine, Faculty of Medicine, University of Tartu, Ravila 19 Street, 50411 Tartu, Estonia

**Keywords:** sexually transmitted infections, sperm functions, male, infertility, *Mycoplasma genitalium*, *Chlamydia trachomatis*

## Abstract

The impact of sexually transmitted infections (STI) on male fertility is controversial. Aims: To investigate the prevalence of urethritis-associated STIs (chlamydia, gonorrhoeae, *Mycoplasma genitalium*, trichomoniasis) among infertile males; to analyze the effect of STIs on semen parameters and blood PSA. Case-control study. Study group (*n* = 2000): males with fertility problems or desire for fertility check. Control group (*n* = 248): male partners of pregnant women. Analyses: polymerase chain reaction for STI, seminal interleukin 6 (IL-6), semen and fractionated urine, blood analyses (PSA, reproductive hormones). The prevalence of *M. genitalium* and chlamydia in the study group was 1.1% and 1.2%, respectively. The prevalence of chlamydia in the control group was 1.6%, while there were no *M. genitalium* cases. No cases with gonorrhoeae or trichomoniasis or combined infections were observed in neither group. There was a higher seminal concentration of neutrophils and IL-6 among *M. genitalium* positives compared with STI negatives. There was a trend toward a lower total count of spermatozoa and progressive motility among STI positives. No impact of STIs on PSA was found. The prevalence of STIs among infertile males is low. *M. genitalium* is associated with seminal inflammation. The impact of STIs on semen parameters deserves further investigations.

## 1. Introduction

Infertility is defined as the inability of a couple to become pregnant despite unprotected intercourse for a period of more than twelve months [1]. The prevalence rate of infertility varies from 3.5% to 16.7% in more developed countries and from 6.9% to 9.3% in less-developed countries, with an estimated overall median prevalence of 9% [2]. In 50% of involuntarily childless couples, a male infertility-associated factor is found usually together with abnormal semen parameters [3].

Urogenital infections and inflammation are accepted contributing factors of male infertility in 6.9 to 16% of men [4,5]. Inflammation and oxidative stress could serve as core mechanisms linking STI with male infertility [6]. The impact of sexually transmitted infections (STI) on male fertility is a controversial topic. According to Ochsendorf et al., depending on the local prevalence of STIs and the availability of medical care, the impact of STIs on male urogenital system and fertility (as a consequence of infection) may appear regionally different [7]. Another problem is the distinction of the impact of STI agents between infection with pathological consequences and contamination without relevance [7]. Inconsistent diagnostic criteria applied to date could also explain the controversy about the role of infection and inflammation in the genital tract as a cause of infertility [8]. For example, using detection methods for *Chlamydia trachomatis* other than nucleic acid amplification tests (NAAT) including culture or serological methods is of significant concern. Culture for *C. trachomatis* detects only viable infectious chlamydial elementary bodies and the sensitivity is at most 70–85%. The major drawback of serological tests for C. trachomatis is that a positive antibody test will not distinguish a previous from a current infection [9].

Ahmadi et al. published a systematic review and meta-analysis on the association of *C. trachomatis* with infertility and clinical manifestations. The authors concluded that urogenital *C. trachomatis* prevalence was significantly higher in the infertile men compared with the fertile men (overall odds ratio 2.2, 95% confidence interval 1.3–3.7) and in symptomatic men compared with asymptomatic men (overall odds ratio 4.9, 95% confidence interval 1.1–21.7). However, the authors guided the attention to the fact that many studies were excluded from the analyses because the detection method was other than NAAT. This may suggest participation bias in the generalization of the meta-analysis results. Secondly, the number of included studies in males in both the fertile-infertile and the symptomatic-asymptomatic groups was fewer than 10 and therefore insufficient for a good meta-analysis and an accurate conclusion [10].

Less is known about the association between male infertility and other STIs. There is only limited information about the role of Trichomonas vaginalis in male infertility [11]. The lack of studies on Neisseria gonorrhoeae is likely due to the rarity of this infection in developed countries [12]. Little work has been carried out on the possible effects of the newest STI-causing agent *Mycoplasma genitalium* on male infertility, too [13,14]. To sum up, the association between STIs and male infertility still needs additional investigations, especially concerning *M. genitalium*.

The aim of the current study was to investigate the prevalence of urethritis-associated STIs (*Chlamydia trachomatis*, *Neisseria gonorrhoeae*, *Mycoplasma genitalium*, and *Trichomonas vaginalis*) among male patients referring to Andrology Centre (Tartu University Hospital, Tartu, Estonia) due to fertility problems. We also aimed to analyze the effect of particular STI agents on semen parameters and blood PSA levels.

## 2. Results

### 2.1. Basic Clinical and Semen Parameters of Study Subjects

The basic parameters of study subjects are presented in Table 1.

Among 2000 infertile men, the proportion of primary and secondary infertility was 74.4% (1487 patients) and 25.6% (513 patients), respectively. The median duration of infertility was 1.9 years with the range 1.0–20.0 years.

The infertility group had lower testicular volume, higher body weight, and a larger proportion of overweight patients (BMI > 24.9) compared to the control group. In addition, infertile men had a lower total count and concentration of spermatozoa, lower percentage of spermatozoa with progressive motility, and normal morphology. The concentration and percentage of neutrophils in semen were also higher among infertile men. However, there was no difference in seminal IL-6 levels between groups. Importantly, there was no difference in abstinence time between infertile and fertile groups.

There was a significantly higher blood level of LH among the infertility group compared to the control group. There was also a slight trend for higher FSH and PSA blood levels among infertile men compared to fertile men, albeit statistically insignificant.

### 2.2. Prevalence of STI among Study Subjects

In the first part of our study, we aimed to describe the prevalence of STIs among fertile and infertile patients.

The prevalence of STIs was 2.3% in the infertility group and 1.6% in the control group. Among the infertility group patients, there were 22 men with *M. genitalium* and 24 men with *C. trachomatis* that comprise 1.1% and 1.2% prevalence, respectively (Table 1). The prevalence of *C. trachomatis* among control group patients was 1.6% while no cases with *M. genitalium* were found. There were no cases with *N. gonorrhoeae, T. vaginalis*, or combined infections in neither group. Interestingly, we found two *C. trachomatis* cases with reported acute orchitis in anamnesis that may suggest reinfection or untreated STI condition.

### 2.3. Impact of STI on Inflammation in Semen

For the second step, we aimed to describe the association of STIs with inflammation in male genital tract.

We made the following analyses of both infertile and fertile groups together because the total number of STI-positive patients among fertile patients was considerably low (only four *C. trachomatis* cases). The association of *M. genitalium* and *C. trachomatis* with semen neutrophils’ concentration and IL–6 concentration is shown in Figure 1A,B, respectively. Both markers were statistically higher for *C. trachomatis* and *M. genitalium*-positive patients compared with STI-negative patients.

We also tried to find the best cut–off value to predict *C. trachomatis* and/or *M. genitalium* infection using semen concentration of neutrophils as a descriptive variable. Using different previously proposed [15,16] cut-off levels for seminal neutrophils’ concentrations, the proportion of STI-positive/negative patients changed but not significantly (Appendix A).

We found the best cut-off value for semen neutrophils’ at a concentration of 0.28 mill/mL for predicting *C. trachomatis* and/or *M. genitalium* infection with sensitivity 58.0%, specificity 71.3%, positive prognostic value 1.3%, negative prognostic value 95.6%, and area under the curve 0.691. For seminal IL-6, the optimal cut-off value predicting presence or absence of *C. trachomatis* and/or *M. genitalium* infection was 17.5 ng/L with sensitivity 75.5%, specificity 47.8%, positive prognostic value 1.2%, negative prognostic value 96.8%, and area under the curve 0.644 (Appendix A).

**Figure 1 ijms-22-13467-f001:**
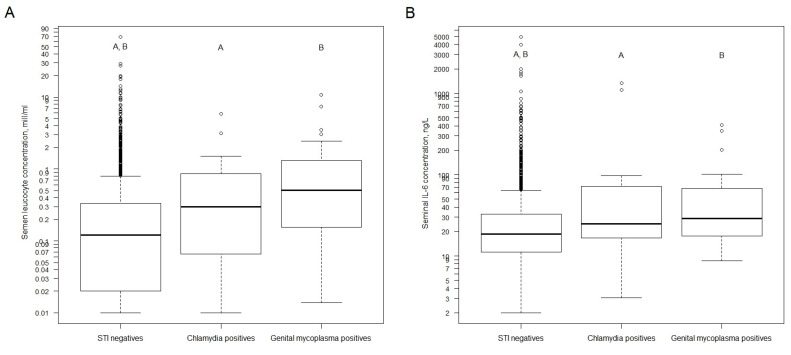
(**A**). Distribution of semen leucocyte concentration by sexually transmitted infection status (both unselected fertile and infertile group, without exclusion of patients for hormonal, testicular volume, and infertility causal factors). (**B**). Distribution of seminal plasma IL-6 concentration by sexually transmitted infection status (both fertile and infertile group, without exclusion of patients for hormonal, testicular volume, and infertility causal factors). Abbreviations: STI—sexually transmitted infections. Data for semen leucocyte concentration are not available for five patients. Data for seminal plasma IL-6 are not available for 65 patients. Mann–Whitney U test with Bonferroni correction for two tests (STI negatives vs. Chlamydia positives, STI negatives vs. Genital mycoplasma positives). For subfigure A: A—*p*-value < 0.001, B—*p*-value = 0.007. For subfigure B: A,B—*p*-value = 0.01. For subfigure B: A,B—*y*-axis is natural logarithm scale.

### 2.4. Impact of STI on Blood PSA and Semen Quality

In the third part of our study, we aimed to reveal the association of particular STI agents with the quality of semen parameters and blood PSA level. The selection of the data was performed to avoid possible confounding before analysis.

To eliminate possible confounding with known factors influencing semen quality and reproductive hormone levels, we excluded 625 male patients from the main study group and 57 male patients from the main control group additionally. The reasons and order of decision to eliminate study subjects additionally were defined according to the following criteria: (I) defined causal factors of male infertility according to Punab et al. [5]: genetic causes, secondary hypogonadism, congenital anomalies: systemic and/or in urogenital tract, serious sexual dysfunctions, oncological diseases, seminal tract obstruction, other testicular factors that include acquired testicular damage (TD) [exposure to high dose radiation in Chernobyl, testis trauma with volume change, mumps orchitis, other orchitis, epididymitis, testicular torsion, hernia operation with ipsilateral TD, epididymal cyst operation with ipsilateral TD, hydrocele operation with ipsilateral TD, other testis operation with ipsilateral TD] and secondary testicular damage [anabolic steroids, medication (salasopyrin, trexan), status diagnosed post kidney transplantation]; (II) bitesticular volume < 30 mL for unexplained reason; (III) hormonal changes: FSH ≥ 8.0 U/L and/or LH > 9.4 U/L and/or FSH < 1.0 U/L and/or LH < 1.0 U/L and/or total testosterone < 10.5 nmol/L).

Detailed information with references for elimination criteria is provided in Appendix A.

After performing the abovementioned exclusion, there remained 1345 STI-negative, 14 *C. trachomatis*-positive and 16 *M. genitalium*-positive patients in the selected infertility group and 188 STI-negative and three *C. trachomatis*-positive patients in the selected control group (Figure 2).

The detailed data about dropped out patients is presented in Appendix A.

In this step of the study, we decided not to combine patients from the selected infertility and control groups to avoid possible bias of the results. As the total number of STI-positive patients in the selected control group was low (only three *C. trachomatis*-positive patients), we performed all the following calculations for the selected infertile group’s patients only.

After performing the abovementioned data correction, we did not reveal any impact of *M. genitalium* and *C. trachomatis* on blood levels of PSA (Appendix A).

The semen parameters of the selected infertility group’s patients according to STI status (Figure 1) are presented in Table 2.

Generally, STI-positive patients (both *C. trachomatis* and *M. genitalium* cases) had significantly lower total counts of spermatozoa and total counts of spermatozoa with progressive motility compared to STI-negative patients (Table 2). The count of spermatozoa with normal morphology was also lower among STI-positive patients, however, the observed difference was over the significance level.

**Table 2 ijms-22-13467-t002:** Impact of *C. trachomatis*, *M. genitalium*, and their combination on semen parameters (selected infertile group only, after exclusion of patients for hormonal, testicular volume, and infertility causal factors).

Parameter	STI Negative *n* = 1345	Either *C. trachomatis* or *M. genitalium* Positive *n* = 30	STI Negative vs. Either *M. genitalium* or *M. genitalium* Positive, *p*-Value ^$$^	*C. trachomatis* Positive *n* = 14	STI Negative vs. *C. trachomatis* Positive, *p*-Value ^$$,^^①^	*M. genitalium* Positive *n* = 16	STI Negative vs. *M. genitalium* Positive, *p*-Value ^$$,^^①^
Abstinence before semen analysis, in days	4.0 (0.0–56.0); 3.0; 4.0	4.0 (1.0–7.0); 3.0; 5.0	0.920	3.5 (2.0–7.0); 2.0; 7.0	0.186	4.0 (1.0–7.0); 2.8; 5.0	0. 191
Volume of seminal fluid, in mL	4.1 (0.8–40.0); 3.0; 5.2	4.1 (1.8–9.0); 3.4; 5.3	0.812	3.7 (1.8–9.0); 3.6; 4.2	0.188	4.4 (1.8–6.9); 3.0; 5.4	0.159
Concentration of spermatozoa, M/mL	56.0 (0.0–440.0); 30.0; 94.5	47.0 (1.3–210.0); 9.1; 69.8	0.034	49.0 (1.3–110.0); 17.1; 51.5	0.164	36.5 (1.3–210.0); 7.1; 95.0	0.396
Total count of spermatozoa, in M	226.2 (0.0–2432.0); 110.4; 372.8	151.0 (2.3–1113.0); 39.6; 303.1	0.028	169.5 (10.1–405.0); 55.9; 201.2	0.132	134.4 (2.3–1113.0); 42.0; 444.8	0.380
Spermatozoa with progressive motility, %	47 (0–83); 35; 57 ^A^	41 (0–71); 23; 57	0.187	48 (0–71); 39; 58	0.167	34 (3–64); 19; 57	0.088
Total count of spermatozoa with progressive motility, in M	101.8 (0.0–1056.0); 44.3; 186.7 ^A^	68.6 (0.0–489.7); 9.3; 156.6	0.034	75.9 (0.0–281.2); 22.8; 113.9	0.356	50.7 (0.3–489.7); 6.4; 202.5	0.194
Spermatozoa with normal morphology, %	8 (0–53); 4; 12 ^B^	5 (0–24); 2; 24	0.242	8 (0–22); 3; 14	0.188	4 (0–24); 2; 10	0.184
Total count of spermatozoa with normal morphology, in M	16.4 (0.0–389.1); 5.3; 37.5 ^B^	7.9 (0.0–189.2); 0.4; 35.5	0.105	14.3 (0.0–69.2); 1.7; 34.6	0.952	4.8 (0.0–189.2); 0.3; 33.3	0.236
Concentration of round cells in seminal fluid, in M/mL	1.6 (0.0–75.0); 0.6; 3.0	1.8 (0.2–31.0); 0.8; 3.9	0.485	1.2 (0.2–7.2); 0.5; 2.0	0.746	3.0 (0.4–31.0); 1.0; 7.3	0.146
Percentage of neutrophils in seminal fluid, %	8 (0–91); 3; 16	22 (0–63); 11; 39	7.379 × 10^−6^	20 (4–48); 13; 30	0.004	25 (0–63); 10; 43	0.002
Concentration of neutrophils in seminal fluid, in M/mL	0.1 (0.0–68.3); 0.0; 0.3	0.2 (0.0–10.70); 0.1; 0.9	0.001	0.2 (0.0–1.4); 0.1; 0.7	0.250	0.4 (0.0–10.7); 0.1; 2.6	0.004
IL-6 in seminal plasma, in ng/L	18.2 (2.0–4928.0); 11.1; 32.3 ^C^	30.4 (3.0–1330.0); 17.9; 78.7 ^D^	0.002	24.8 (3.0–1330.0); 20.8; 66.2 ^D^	0.192	30.6 (9.2–406.0); 17.6; 84.5	0.016

Annotations to this table. Data are presented as median (range), 25th, and 75th percentile. Data are with non-parametric distribution. **^$$^** *p*-value is calculated with Mann–Whitney test comparing two groups. ^①^ Bonferroni correction of *p*-value for two tests. ^A^ Data accounting progressive motility of spermatozoa are not available for one patient. ^B^ Data accounting spermatozoa morphology are not available for one patient. ^C^ Data accounting IL-6 in seminal plasma are not available for 36 patients. ^D^ Data accounting IL-6 in seminal plasma are not available for one patient.

However, when STI-positive cases were analyzed separately, the impact of *C. trachomatis* on the total count of spermatozoa and progressive motility became slightly over the significance level. Similar tendencies were also seen in the case of *M. genitalium*-positive patients. The spermatozoa count with normal morphology was lower in *M. genitalium*-positive patients though the difference between the groups was slightly over the significance level again.

The impact of both combined *C. trachomatis* and *M. genitalium* on seminal inflammatory parameters (concentration of neutrophils, seminal IL-6) also remained significant in the selected infertility group patients. However, when STI-positive cases were split separately, the impact of *C. trachomatis* on seminal IL-6 and neutrophil concentration disappeared, while remaining for *M. genitalium* (Table 2).

## 3. Discussion

We found a low prevalence of STIs among both fertile (1.6%) and infertile (2.3%) men in Estonia. *C. trachomatis* prevalence was 1.2% among infertile men and 1.6% among the fertile control group. At the same time, *M. genitalium* prevalence was 1.1% among infertile but we did not find any case among the fertile control group. There were no cases with *N. gonorrhoeae* or *T. vaginalis* in either group. *M. genitalium* did provoke an inflammatory reaction in semen, but in most cases far below the WHOs proposed limits. STI-positive patients had a significantly lower total count of spermatozoa and total count of spermatozoa with progressive motility compared to STI-negative subjects.

### 3.1. Advantages and Disadvantages of the Study

Our study had some major advantages compared to previous ones. First, we included the control group of fertile men. Secondly, to analyze the influence of STIs on semen parameters we formed a group of clearly defined idiopathic infertile cases where all related diseases, anatomical, and hormonal factors with potential effect or relation to semen parameters were filtered out. Thirdly, the diagnosis of STI was made with nucleic acid amplification tests. The sensitivity and specificity of NAATs are far better than that of any of the other tests used in clinical practice [17].

A major disadvantage of our study is the relatively small groups of *C. trachomatis* and *M. genitalium*-positive patients. At the same time, this indicates a beneficial epidemiological situation of STIs in our country and points out to target other groups with higher prevalence of STIs (i.e., young sexually active men) to disentangle the multifactorial impact of particular infective agents on semen parameters. At the same time, the groups with fertility issues could serve as acceptable controls.

### 3.2. Epidemiologic Issues

We screened study subjects for four causative agents of STIs (*N. gonorrhoeae, C. trachomatis, M. genitalium,* and *T. vaginalis*), while found only two of them. This is (more or less) in accordance with the overall STI pattern in developed countries where *C. trachomatis* and *M. genitalium* tend to be more frequent urethritis-associated pathogens than *T. vaginalis* and *N. gonorrhoeae*. On the other hand, substantial differences exist in published STIs’ prevalence among the infertile men.

Both *C. trachomatis* and *M. genitalium* are atypical bacteria that cannot be cultured using routine methods. *C. trachomatis* is an obligate intracellular Gram-negative bacterium surrounded by a rigid cell wall. This bacterium has a unique life cycle with two morphologically distinct life stages, elementary bodies, and reticulate bodies. *M. genitalium* is a small slowly growing facultative intracellular bacterium lacking a cell wall around its cell membrane and having the smallest genome (~580 kb) for a self-replicating organism [13]. The prevalence of *C. trachomatis* among infertile men varies depending on the place of the study and the year of the publication. Early studies have shown higher, while recent studies lower prevalence of chlamydia. This could suggest the positive effect of preventive work in the field of sexual medicine in developed countries during the last decades.

In comparison with other studies, our prevalence estimates for *C. trachomatis* among infertile and fertile men were lower than in the USA (39.3%) [18], Czech Republic (13–21.7%) [19], Germany (10%) [20], and France (10.9%) [21].

While there are plenty of studies analyzing the role of *C. trachomatis* on male fertility, there is a scarce number of works about *M. genitalium* so far. *M. genitalium* prevalence among infertile men in our study was similar to the studies from Denmark (0.9%) [22] and Croatia (1.4%) [23]. The recent meta-analysis does not support the etiological role in male infertility for *M. genitalium* [14]. However, there were included only three original studies regarding this bacterium and, therefore, the analysis gave borderline significance value, despite good odds ratio [3.27 (95% CI: 0.80–13.29)].

### 3.3. Impact of STI on Seminal Inflammation: Seminal Leukocytes

We found higher semen leucocyte concentration among STI-positive patients compared with STI-negative patients. This difference arose mostly from the group of *M. genitalium*-positive cases, while among *C. trachomatis* cases there is only a visible elevated percentage of neutrophils among round cells in semen.

Our results showing a positive association between *M. genitalium* and leucocytospermia are contradictory to the study by Kjaergaard et al. who did not find such an association [22]. The possibility of partial (methodical) art-fact in the context of this study cannot be excluded (e.g., *M. genitalium* detection by in-house PCR, detection of *C. trachomatis*, and *T. vaginalis* by culture methods).

Our and some other studies [22,24] do not support the association of *C. trachomatis* with leucocytospermia. The only observed significant finding in *C. trachomatis*-positive men in our study was a higher percentage of neutrophils in seminal fluid among selected infertile group patients, but there was no significance for the concentration of neutrophils in seminal fluid. The percentage of neutrophils in seminal fluid is a ratio parameter and there is no officially accepted reference level for that assessing seminal inflammatory process. For this reason, we cannot use this parameter for a reliable investigation of seminal inflammation. However, other researchers found an association between leucocytospermia and *C. trachomatis* infection [25]. One explanation for the observed differences in the association of chlamydia with pyospermia across different studies could be the inability of studies to discriminate between recent and old chlamydia infection because a large proportion of *C. trachomatis*-positive patients can be asymptomatic. The addition of systemic serological tests into diagnostic armamentarium is also not useful, because antibodies elicited by *C. trachomatis* are long lived and a positive antibody test will not distinguish a previous from a current infection [8].

Based on our study results, we propose the optimal cut-off level for predicting *M. genitalium* and/or *C. trachomatis* infection to be 0.28 million leukocytes per ml. This value is lower than 1 million/mL proposed by the WHO [26]. The suggestion to lower the cut–off level for leucocytospermia was also supported by our previous study [15] as well as by some other studies [16].

### 3.4. Impact of STI on Seminal Inflammation: Seminal Interleukin-6

Besides neutrophils’ count, we also assessed seminal interleukin-6 as a potential additional inflammatory marker in our study. In some previous studies, seminal IL-6 is positively correlated with seminal WBCs [27,28]. We observed significantly higher IL-6 levels for *M. genitalium*-positive patients compared with STI-negative patients. We did not find any human studies reporting the association between seminal IL-6 and *M. genitalium*. Therefore, our study is the first to report such an association. To the best of our knowledge, we found only two studies that observed the relationship between seminal IL-6 and *C. trachomatis* in men. Both studies found elevated IL-6 in *C. trachomatis*-positive men [29,30].

### 3.5. Impact of STI on Seminal Inflammation: PSA

Studying the impact of STIs on prostate inflammation is hindered by two major problems: (1) diagnostic material passing the urethra may reflect only urethral contamination; (2) prostatic biopsy specimens from the gland may also contain cutaneous, intestinal, or urethral material [31]. Korrovits et al. showed that seminal IL-6 and serum PSA are excellent negative predictive markers for asymptomatic inflammatory prostatitis in young men [32]. We did not find any study evaluating the impact of *M. genitalium* on PSA. Motrich et al. did not find any impact of *C. trachomatis* on the PSA level [33].

The impact of *C. trachomatis* and *M. genitalium* on PSA could serve as a surrogate marker for studying causality for the impact of STIs on prostate inflammation. However, our attempt to use the PSA to that end failed. The possible explanation for that is the large age range of patients (18–50 years), elusive temporal factor (it is virtually impossible to distinguish between recent and old infection because a large proportion of cases are asymptomatic; time frames between urethral and prostatic infection, prostatic infection and blood PSA response), high variability in the PSA value (partly because of large age range) and possibly, the factor of prostate volume (higher PSA values in patients with larger prostate), and the low number of STI-positive patients. Due to that, thorough statistical adjustment with enough power was impossible. Some researchers associated *M. genitalium* with chronic prostatitis [34,35,36]. Hence, this topic requires further detailed investigations addressing possible caveats listed earlier in this paragraph.

### 3.6. Impact of STI on Semen Quality and Particularly on Sperm Count

We found a significantly lower total count of spermatozoa and total count of spermatozoa with progressive motility among STI-positive subjects.

In vitro studies have confirmed the impact of chlamydia on sperm cells while in vivo studies are controversial [37]. In our study, we found biologically meaningfully but not statistically significantly lower counts of spermatozoa, but no effect on sperms’ progressive motility. Previous studies [20,24,25,29,33,38] also do not support the impact of *C. trachomatis* on sperm motility. On the other hand, there are different studies supporting the impact of *C. trachomatis* on sperm concentration, motility, and morphology [18,19].

In our study, there was also a clear trend toward a lower concentration, total count of spermatozoa, % of spermatozoa with progressive motility, and also % of sperms with normal morphology in *M. genitalium*-positive patients, though the observed difference between the groups was once again slightly over the significance level. Perhaps, the reason for this is a relatively small number of *M. genitalium* patients, inherently high variability of semen parameters, and decreased power of statistical tests. The impact of *M. genitalium* on semen parameters was assessed in previous studies and the results tend to be controversial, too. It is worth mentioning that, to date, evidence for *M. genitalium* is quite scarce. There are only a few studies from the Western world addressing the impact of *M. genitalium* on semen parameters. Some studies have revealed no impact of *M. genitalium* on semen parameters [22,38] while Yan et al. reported the negative impact of *M. genitalium* on progressive motility of spermatozoa [39].

The abovementioned data and the fact that we found *M. genitalium* cases only among infertility group patients suggests the role of this bacteria in male impaired fertility and deserves further investigations in larger cohorts or other study populations where the prevalence of STIs is supposed to be higher (i.e., young sexually active men).

## 4. Materials and Methods

### 4.1. Forming the Study Group—Infertile Couples’ Males

The study was approved by the Ethics Review Committee on Human Research of the University of Tartu, Estonia (permission 152/4 (last amendment 288/M-13) and 188/M-16 (control group)). The study was conducted according to the Declaration of Helsinki principles. Written informed consent was obtained from patients prior to recruitment.

This was a retrospective analysis of the prospective cohort’s data. During the period January–December 2012, there were 3095 referrals to Andrology Centre (Tartu University Hospital, Tartu, Estonia) due to family fertility problems. Among 3095 referrals, there were 38 repeated cases, which were eliminated. Study subjects were included without regard to their semen quality. Among these 3057 men, 1034 non-participants did not count the current part of the study important, as their partners already passed STI tests during their infertility workup or just forgot to deliver the required material. In addition, 23 patients were eliminated from the study group due to the following reasons: age over 50 years (22 patients), and incorrectly inserted information about a patient (one patient). The final study group consisted of 2000 men who fulfilled inclusion criteria (age 18–49 years; correctly delivered first-voided morning urine to the laboratory). In the second stage of the study, we excluded 625 male patients from the main study group with known factors influencing semen quality and reproductive hormone levels to eliminate possible confounding (see Appendix A). The selected final study group comprised 1375 men. The formation of the study group is shown in Figure 2.

### 4.2. Forming the Control Group—Partners of Pregnant Women

The control group ‘Partners of pregnant women’ represented a reference sample of fertile control men (Figure 1). The control group’s composition is detailed in a recent publication [40]. Briefly, in 2010–2012, male partners of informed pregnant women at the Women’s Clinics of Tartu University Hospital and West–Tallinn Central Hospital were invited to participate in the study and approximately 30% of eligible men agreed. The participants had a choice to complete solely a structured medical questionnaire or to pass additionally a standard andrological physical examination along with blood hormone, semen analysis and first-voided urine sample for urethral inflammation detection and STI tests. Among 277 participants who comprised the control group, 29 patients were excluded for the following reasons: 20 patients had not performed the required STI tests, six patients were older than 50 years, and three patients used assisted reproductive techniques. Thus, the full dataset including completed questionnaire and physical examination, STI tests, semen analysis (identical to that of the infertility patients) was collected for 248 men who comprised the final control group. To reveal the association of a particular STI agent with the quality of semen parameters and blood PSA level in the second stage of the study, we excluded additionally 57 men from the final control group (see Appendix A). The selected final control group consisted of 191 men.

The formation of the control group is shown in Figure 1.

### 4.3. Clinical Examination

Patients were examined by six clinicians, who had received the respective training in clinical assessment and standardized andrological workup, locally and in collaboration with the other centers accredited by the European Academy of Andrology. Overall, 59.9% and 19.8% of all study group subjects were investigated by two senior clinicians, MP and OP, respectively. The remaining patients (20.3%) were examined by four clinicians. Control group subjects were investigated by MP and KP. An assisting nurse recorded the subject’s height and weight. Physical examination for the assessment of genital pathology and testicular size was performed with the man in the standing position. If necessary, pathologies were clarified further with the men in a supine position. The orchidometer (made of birch wood, Pharmacia & Upjohn, Denmark) was used for the assessment of testicular size. The total testes volume was the sum of right and left testicles. The position of the testicles in the scrotum, pathologies of the genital ducts (*epididymis* and *ductus deferens*) and the penis, presence and grade of varicocele were registered for each study participant. Varicocele was graded according to a traditional system as follows: Grade 1—palpated only on the Valsalva maneuver; Grade 2—venous distension easily palpable but not visible; Grade 3—venous plexus bulges through the scrotal skin, visible and palpable. Varicoceles were classified according to the highest assigned grade, independent from the affected side. Objective physical examination and interviewing on the patient’s medical history were applied to diagnose existing and retrospective cases of cryptorchidism and document past operations due to inguinal hernia.

### 4.4. Laboratory Analyses

#### 4.4.1. First-Voided Urine

Patients were asked to provide 10–15 mL of first-voided urine (FVU) representing the urethral washout. FVU was collected into pre-weighted aseptic 150 mL urine container. Patients were told to abstain from washing genitalia before analyses. The minimal accepted time since the last micturition until analysis was four hours. FVU was analyzed by dipstick test. The dipstick test was performed using Combur10Test UX test strips (Roche Diagnostics Ltd., Rotkreuz, Switzerland) and Urisys 1100 machine (Roche Diagnostics GmbH, Mannheim, Germany) according to the manufacturer’s instruction. Urine was considered inflammatory if there were ≥25 white blood cells per one milliliter (WBC/mL).

#### 4.4.2. Sexually Transmitted Infections

FVU was used for the detection of STIs. STIs were detected from first-voided urine using the polymerase chain reaction (PCR) method (*C. trachomatis* and *N. gonorrhoeae* DNA by Roche/Cobas^®^ 4800 CT/NG Test [Roche Diagnostics Ltd., Rotkreuz, Switzerland]; *M. genitalium* DNA by Sacace Biotehnologies/*Mycoplasma genitalium* Real-TM [Sacace Biotechnologies Srl, Como, Italy]; *T. vaginalis* DNA by Sacace Biotehnologies/Trichomonas vaginalis Real-TM [Sacace Biotechnologies Srl, Como, Italy]) according to the manufacturer’s instructions, at the United Laboratory of Tartu University Hospital. All STI-positive patients were contacted for the appropriate treatment and follow-up.

#### 4.4.3. Semen Analysis

Semen samples were obtained by patient masturbation and all semen values were determined in accordance with the World Health Organization (WHO) recommendations at the time of recruitment [26]. In brief, subjects were told to abstain from ejaculation for three to four days. After ejaculation, semen was incubated at 37 °C for 30–40 min for liquefaction. The semen volume was estimated by weighing the collection tube with the semen sample and subsequently subtracting the predetermined weight of the empty tube and assuming 1 g = 1 mL. For the assessment of the spermatozoa concentration, the samples were diluted in a solution of 0.6 mol/L NaHCO_3_ and 0.4% (*v/v*) formaldehyde in distilled water. The spermatozoa concentration was assessed using the improved Neubauer hemocytometer. Motility was assessed in order to report the number of progressively motile spermatozoa. Smears for the morphology assessment were made. Following the fixation and Papanicolaou staining, the morphology was assessed according to strict criteria [26].

For detection of the inflammation, the white blood cells (WBC) count and interleukin-6 (IL-6) level in semen were measured. Semen smears were made for the detection of WBCs. The smears were air-dried, Bryan–Leishman stained, and examined with the use of oil immersion microscopy (magnification: 1000×) by an experienced microscopist. Polymorphonuclear leukocytes were differentiated from spermatids by the presence of segmented nuclei, bridges between lobes of a nucleus, and the specific granulation of the cytoplasm. The WBC concentration in the semen was calculated by using the known spermatozoa concentration. In cases of low semen volume (<1.5 mL) and in clinical cases experiencing an orgasm with missing antegrade ejaculation, the retrograde ejaculation was confirmed by examining a sample of post-ejaculatory urine for the presence of spermatozoa. IL-6 levels in the seminal plasma were measured using the Immulite automated chemiluminescence immunoassay analyzer (Immulite DPC, Los Angeles, CA, USA) according to the manufacturer’s instructions at the United Laboratories of Tartu University Hospital. If the seminal secretion volume was <20 μL, a known volume of plasma was diluted according to the manufacturer’s instructions. Any dilution factor was recorded and results were corrected accordingly.

#### 4.4.4. Blood Assays

Venous blood of the patient was drawn from the cubital vein in the morning from 08:00 a.m. to 10:30 a.m. and serum was separated immediately. Follicle-stimulating hormone (FSH), luteinizing hormone (LH), total testosterone, and total prostate-specific antigen (PSA) levels of blood serum were measured using the Immulite automated chemiluminescence immunoassay analyzer (Immulite; Diagnostic Products Corp., Los Angeles, CA, USA) according to the manufacturer’s instructions, at the United Laboratories of Tartu University Hospital.

#### 4.4.5. Genetic Analyses

Tests for the known genetic causes of male infertility (karyotyping, Y–chromosome microdeletions [*AZFa, AZFb, AZFc*], *CFTR* mutations [p.F508del, 394delTT, IVS8 5T/7T/9T]) were performed at the United Laboratories of Tartu University Hospital. A detailed description of genetic analyses is provided elsewhere [5].

### 4.5. Data Processing and Statistical Analysis

Four dedicated study nurses (two in each department—Tallinn and Tartu) entered the collected epidemiological, laboratory, and clinical examination data into two separate, but identically structured, databases. Prior to statistical analysis, the two databases were merged and duplicate entries were eliminated. The entered laboratory data were counter–controlled from primary sources (lab databases) and, if needed, edited by a specially trained researcher (MP and KP). Clinical data relevant to defining the cause of male infertility were controlled retrospectively for all study subjects from their medical case histories one by one by one author of the study (MP).

Statistical analyses were performed using Microsoft Excel software (Microsoft Corporation, Washingron, USA) and RStudio software (R version 3.6.1. (2019–07–05), RStudio Inc., Massachusetts, USA). For comparison of numerical continuous parameters between different STI groups, the non-parametric Mann–Whitney *U*-test was used for non-normally distributed data, while the parametric unpaired *t*-test was used for normally distributed data. For categorical nominal data, Fisher’s exact test and chi-squared test were used. Two-sided *p*-value ˂ 0.05 was accepted as statistically significant. Comparing *C. trachomatis*-positive vs. STI-negative and *M. genitalium*-positive vs. STI-negative patients, the Bonferroni correction of *p*-values for two tests was used. To find the best cut-off value for semen neutrophils’ concentration and semen interleukin-6 for predicting *C. trachomatis* and/or *M. genitalium* infection, the construction of receiver operating curves (ROC) was performed. Due to the relatively small number of STI-positive patients, the construction of multivariate logistic regression models was avoided.

## 5. Conclusions

The prevalence of urethritis-associated STI among infertile patients in Estonia is low, only two out of four tested causative agents were found in them—*M. genitalium* 1.1% and *C. trachomatis* 1.2%. Markers of inflammation displayed higher levels in the semen of *M. genitalium*-positive men while currently accepted seminal leucocytes’ threshold proposed by the WHO does not allow for reliable prediction of the presence of urethritis-associated STIs. Patients infected with *M. genitalium* and *C. trachomatis* had a lower total count of spermatozoa and total count of spermatozoa with progressive motility compared to STI-negative subjects, therefore, the impact of these atypical bacteria on male fertility cannot be ruled out. However, the more precise effect of these bacteria separately still requires further investigation.

## Figures and Tables

**Figure 2 ijms-22-13467-f002:**
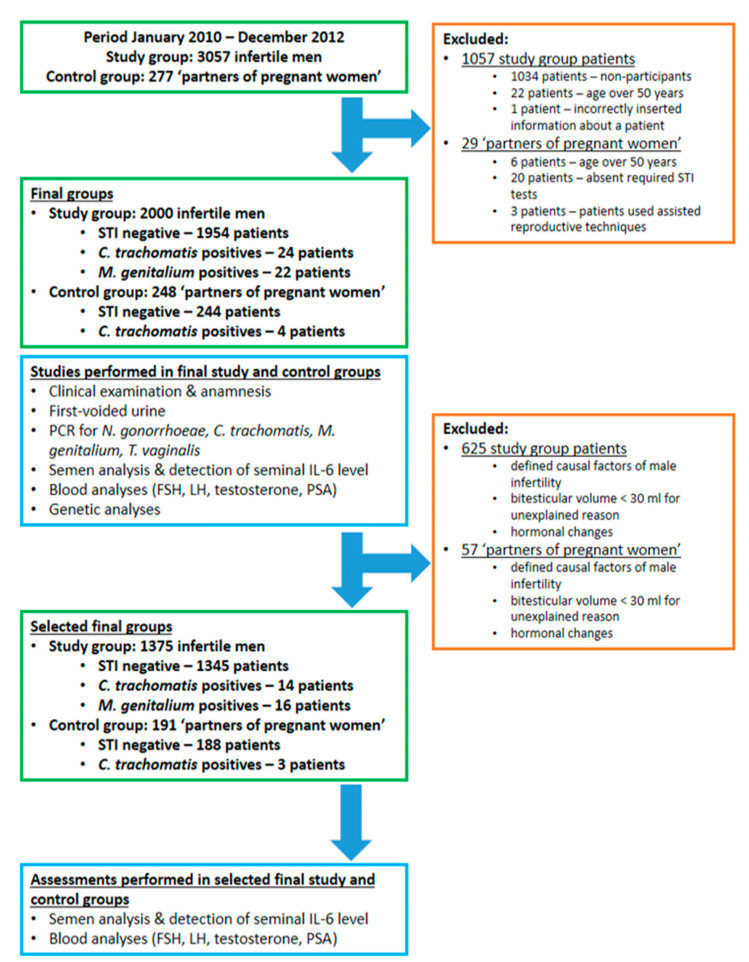
Description of the study protocol. Abbreviations: STI—sexually transmitted infections.

**Table 1 ijms-22-13467-t001:** Basic and semen parameters of the unselected study and control groups (without exclusion of patients for hormonal, testicular volume, and infertility causal factors).

Parameter	Study Group, Infertile Men (*n* = 2000)	Control Group, Fertile Men (*n* = 248)	*p*-Value
	Median (Range); 25th Centile; 75th Centile	Median (Range); 25th Centile; 75th Centile	
Basic parameters			
Age in years	32.5 (18.2–49.8); 32.3 to 32.8 ^$^	31.8 (20.0–50.0); 31.1 to 32.6 ^$^	0.081 ^###^
STI status			
→ Patients without STI	1954 (97.7%)	244 (98.4%)	0.649 ^##^
→ *C. trachomatis* and *M. genitalium* positive patients together	46 (2.3%)	4 (1.6%)	0.649 ^##^
→ *C. trachomatis* positive patients	24 (1.2%)	4 (1.6%)	0.541 ^##^
→ *M. genitalium* positive patients	22 (1.1%)	0 (0.0%)	0.162 ^##^
Bitesticular volume, in mL-s	45.0 (0.0–100.0); 44.6 to 45.5 ^$,A^	47.3 (23.0–100.0); 46.1 to 48.5 ^$,A^	<0.001 ^###^
→ Left testicle’s volume, in mL-s	22.0 (0.0–50.0); 21.8 to 22.3 ^$,A^	22.9 (0.0–50.0); 22.2 to 23.6 ^$,A^	0.016 ^###^
→ Right testicle’s volume, in mL-s	23.0 (0.0–50.0); 22.8 to 23.3 ^$,A^	24.4 (6.0–50.0); 23.7 to 25.1 ^$,A^	<0.001 ^###^
STI in anamnesis	459 (23.0%)	51 (20.6%)	0.422 ^##^
→ *N. gonorrhoeae* in anamnesis	95 (4.8%)	7 (2.8%)	0.197 ^##^
→ *C. trachomatis* in anamnesis	259 (13.0%)	27 (10.9%)	0.419 ^##^
→ *T. vaginalis* in anamnesis	85 (4.2%)	9 (3.6%)	0.739 ^##^
→ *Lues* in anamnesis	15 (0.8%)	1 (0.4%)	1 ^##^
→ HSV in anamnesis	47 (2.4%)	12 (4.8%)	0.032 ^##^
→ HPV in anamnesis	68 (3.4%)	11 (4.4%)	0.364 ^##^
Patients with varicocele	467 (23.4%)	61 (24.6%)	0.691 ^##^
→ Left-sided grade 1 varicocele–	173 (8.7%)	25 (10.1%)	0.668 ^##^
→ Left-sided grade 2 varicocele	211 (10.6%)	27 (10.9%)	0.777 ^##^
→ Left-sided grade 3 varicocele	32 (1.6%)	4 (1.6%)	1 ^##^
→ Right-sided varicocele	5 (0.3%)	0 (0.0%)	1 ^##^
→ Bilateral varicocele	24 (1.2%)	3 (1.2%)	1 ^##^
→ Operated varicocele	22 (1.1%)	2 (0.8%)	1 ^##^
Smokers	667 (33.4%)	74 (31.1%)	0.513 ^##^
Concentration of FSH in serum, in U/L	3.8 (0.1–74.1); 2.6; 5.6 ^B^	3.6 (0.6–15.2); 2.6; 5.1	0.065 ^#^
Concentration of LH in serum, in U/L	3.9 (0.1–25.9); 2.8; 5.4 ^C^	3.6 (0.6–11.7); 2.4; 4.6 ^C^	<0.001 ^#^
Concentration of total testosterone in serum, in nmol/L	16.0 (0.7–47.8); 12.4; 20.4 ^D^	16.6 (6.0–49.3); 12.9; 20.2	0.489 ^#^
PSA in serum, in µg/L	0.71 (0.11–8.34); 0.52; 0.99 ^E^	0.69 (0.14–7.58); 0.47; 0.96 ^E^	0.053 ^#^
Height, cm	181.1 (153.0–209.0); 180.8 to 181.4 ^$, F^	180.8 (167.0–198.0); 180.0 to 181.5 ^$^	0.468 ^###^
Weight, kg	85.1 (47.0–189.0); 75.7; 95.5 ^G^	82.0 (52.0–139.0); 74.5; 90.9	0.005 ^#^
BMI	26.0 (14.9–76.2); 23.6; 28.7 ^H^	24.9 (18.0–42.4); 22.9; 27.8	0.003 ^#^
→ Overweighed patients (BMI > 24.9)	812 (42.3%) ^H^	85 (34.3%)	0.016 ^##^
→ Obese patients (BMI > 29.9)	340 (17.7%) ^H^	34 (13.7%)	0.129 ^##^
Waist circumference, cm	93.0 (62.0–152.0); 85.0; 101.0 ^I^	90.0 (68.5–124.0); 83.5; 97.1 ^I^	<0.001 ^#^
Semen parameters			
Abstinence before semen analysis, in days	4.0 (0.0–56.0); 3.0; 4.0 ^J^	3.0 (1.0–60.0); 3.0; 5.0	0.238 ^#^
Volume of seminal fluid, in mL-s	4.0 (0.0–40.0); 2.9; 5.1 ^J^	3.8 (1.2–9.5); 2.9; 5.1	0.361 ^#^
Concentration of spermatozoa, in M/mL	50.0 (0.0–487.0); 21.9; 88.0 ^J^	79.0 (6.0–355.0); 46.8; 125.0	<0.001 ^#^
Total count of spermatozoa, in M	189.0 (0.0–2432.0); 79.2; 341.3 ^J^	302.5 (14.4–1657.6); 164.8; 499.3	<0.001 ^#^
Spermatozoa with progressive motility, %	45 (0–83); 34; 56 ^K^	53 (11–84); 44; 60	<0.001 ^#^
Spermatozoa with normal morphology, %	7 (0–53); 4; 11 ^L^	10 (0–27); 7; 14	<0.001 ^#^
Concentration of round cells in seminal fluid, in M/mL	1.5 (0.0–75.0); 0.6; 3.0 ^M^	1.5 (0.0–45.0); 0.5; 3.0	0.231 ^#^
Percentage of neutrophils in seminal fluid, %	9.0 (0.0–92.0); 3.0; 18.0 ^M^	4.0 (0.0–69.0); 0.0; 12	<0.001 ^#^
Concentration of neutrophils in seminal fluid, in M/mL	0.1 (0.0–68.3); 0.0; 0.4 ^M^	0.0 (0.0–27.5); 0.0; 0.3	<0.001 ^#^
IL-6 in seminal plasma, in ng/L	18.6 (2.0–4928.0); 11.2; 32.9 ^N^	19.3 (2.0–486.0); 12.0; 35.0	0.676 ^#^

Annotations to this table. Data with non-parametric distribution. ^$^ The only variables with parametric distribution were age, bitesticular volume, left testicle’s volume, right testicle’s volume, and height. For these parameters, the mean, range and 95% confidence interval are presented. ^A^ Testicular volume was not measured for 146 and 3 patients in the study and the control group, respectively; ^B^ Data not available for 52 patients in the study group only; ^C^ Data not available for 51 and 1 patients in the study and the control group, respectively; ^D^ Data not available for 43 patients in the study group only; ^E^ Data not available for 72 and 16 patients in the study and the control group, respectively; ^F^ Data not available for 91 patients in the study group only; ^G^ Data not available for 94 patients in the study group only; ^H^ Data not available for 80 patients in the study group only; ^I^ Data not available for 183 and 4 patients in the study and the control group, respectively; ^J^ Data not available for 4 patients with serious sexual dysfunctions in the study group only; ^K^ Data not available for 73 patients in the study group only; ^L^ Data not available for 74 patients in the study group only; ^M^ Data not available for 5 patients in the study group only; ^N^ Data not available for 65 patients in the study group only; ^#^ Mann–Whitney U test; ^##^ Fisher’s exact test; ^###^ Unpaired *t*–test.

## Data Availability

The data presented in this study are not publicly available due to privacy and ethical reasons.

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
