# Peer review of "Mycoplasma genitalium Provokes Seminal Inflammation among Infertile Males"

_ijms, 2021, doi:10.3390/ijms222413467_

Round 1

Reviewer 1 Report

  1.  The control group is very small and I recommend to consider revison the results and present a comparison between the parameters of infected and uninfected patients, due to the small number of infected screens in general.
  2. I recommend to include U. parvum and U.urealitutsum as infectious agents in the study.

Author Response

Dear Reviewer 1,

Thank you very much for your valuable commentaries and suggestions. I totally agree that the statistical power of the results could be better if we could form the study and the control group with equal number of study subjects. Unfortunately, we could not bring this to life due to the following reasons: 1) men, especially healthy men, are less lean towards medical studies compared with women, 2) it is quite difficult to form the representative control group for studies dealing with fertility issues.

Because the control group was smaller than the study group, and the absolute number of STI positive men among control group was very small (four men in the control group vs. 46 men in the control group) we did not make the full analysis where we could compare all the four different groups of study subjects (i.e. STI positive [control group] vs. STI negative [control group] vs. STI positive [study group] vs. STI negative [study group]). Another important factor is that combination of both groups (i.e. control group and study group) with forward reassortment according to the STI status seemed to us questionable and suspicious because this would introduce the obvious selection bias – it is impossible to mix the semen analyses of fertile and infertile men together as in up to 60% the cause of male infertility remains unexplained (Punab et al, 2017).

These were the main reasons why we decided to do multiple analyses, including the comparison of semen parameters according to STI status in the selected study group (infertile men) only, after exclusion of patients for hormonal, testicular volume and infertility causal factors. These results are presented in Table 2 in the manuscript. STI positive patients (both C. trachomatis and M. genitalium cases) had significantly lower total count of spermatozoa and total count of spermatozoa with progressive motility compared to STI negative patients. When STI positive cases were analyzed separately, the impact of C. trachomatis on total count of spermatozoa and progressive motility became slightly over significance level. Similar tendencies were also seen in case of M. genitalium positive patients. Count of spermatozoa with normal morphology was lower in M. genitalium positive patients though difference between the groups was slightly over the significance level again. M. genitalium had a statistically significant influence on seminal inflammatory parameters (concentration of neutrophils, seminal IL–6).

I do totally agree that the impact of U. urealyticum and U. parvum on male fertility and semen parameters deserves thorough studying. Unfortunately, we were unable to include the data about Ureaplasmas into the present article, because not all the men in the study and control group had the PCR tests for these bacteria at the moment of the study as Ureaplasmas are not universally acknowledged STI agents. Partially performed PCR tests for detection of Ureaplasmas could complicate the analyses of the whole study groups in the current article. At present, our research group is dealing with U. urealyticum and U. parvum issue and we are planning to prepare a separate scientific publication on this topic that is based on another dataset.

With best regards,

Stanislav Tjagur

References:

Punab, M., Poolamets, O., Paju, P., Vihljajev, V., Pomm, K., Ladva, R., Korrovits, P., & Laan, M. (2017). Causes of male infertility: a 9-year prospective monocentre study on 1737 patients with reduced total sperm counts. Human reproduction (Oxford, England)32(1), 18–31. https://doi.org/10.1093/humrep/dew284

Reviewer 2 Report

I enjoyed reading this manuscript, because in my opinion it is well designed study, with high number of included patience. I also agree with advantages and and disadvantages, which authors listed. I only have one question. I didn't noticed anywhere the approval of the ethical committee. Please add this. Otherwise I have no other comments. 

Author Response

Dear Reviewer 2,

Thank you for the kind words and your suggestion.

We do have the approval of the ethical committee. This information is available on line 527 in the revised manuscript. The location of the approval of the ethical committee is placed at the end of the manuscript according to the requirements of The International Journal of Molecular Sciences (see: https://www.mdpi.com/journal/ijms/instructions). I have also added the approval of ethical committee into the main text of the revised version of the manuscript (see lines 348–351).

With best regards,

Stanislav Tjagur

Round 2

Reviewer 1 Report

yes

Author Response

Dear Reviewer 1,

Thank you for the reply. 

As I did not received any further specific comments, no additional changes to the manuscript were done.

With best regards,

Stanislav Tjagur

Reviewer 2 Report

Sorry, I overlooked. I have no further comments.

Author Response

Dear Reviewer 2,

Thank you for the reply.

With best regards,

Stanislav Tjagur